# The indirect health effects of malaria estimated from health advantages of the sickle cell trait

Sophie Uyoga[1], Alex W. Macharia[1], Carolyne M. Ndila[1], Gideon Nyutu[1], Mohammed Shebe[1], Kennedy O. Awuondo[1], Neema Mturi[1], Norbert Peshu[1], Benjamin Tsofa[1], J. Anthony G. Scott [1,2], Kathryn Maitland[1,3] & Thomas N. Williams [1,3]

Most estimates of the burden of malaria are based on its direct impacts; however, its true burden is likely to be greater because of its wider effects on overall health. Here we estimate the indirect impact of malaria on children's health in a case-control study, using the sickle cell trait (HbAS), a condition associated with a high degree of specific malaria resistance, as a proxy indicator for an effective intervention. We estimate the odds ratios for HbAS among cases (all children admitted to Kilifi County Hospital during 2000–2004) versus community controls. As expected, HbAS protects strongly against malaria admissions (aOR 0.26; 95%CI 0.22–0.31), but it also protects against other syndromes, including neonatal conditions (aOR 0.79; 0.67–0.93), bacteraemia (aOR 0.69; 0.54–0.88) and severe malnutrition (aOR 0.67; 0.55–0.83). The wider health impacts of malaria should be considered when estimating the potential added benefits of effective malaria interventions.

[1] Department of Epidemiology and Demography, KEMRI/Wellcome Trust Research Programme, PO Box 230, Kilifi 80108, Kenya. [2] Department of Infectious Disease Epidemiology, London School of Hygiene and Tropical Medicine, London WC1E 7HT, UK. [3] Department of Medicine, Imperial College, St Mary's Hospital, London W21NY, UK. Correspondence and requests for materials should be addressed to T.N.W. (email: tom.williams@imperial.ac.uk)

*P*lasmodium falciparum malaria is a major cause of child morbidity and mortality in tropical and sub-tropical regions of the world. While a substantial proportion of this mortality is directly attributable to the complications of individual infections[1], malaria may also have a wider impact through its indirect effects on children's health[2,3]. For example, malaria is a recognized cause of malnutrition[4], itself a major determinant of disease susceptibility[5], and can also predispose children to invasive bacterial infections[6]. Nevertheless, quantifying this indirect effect can be difficult[7]. Intervention studies can be hard to interpret because most interventions are not entirely specific to malaria and can result in wider impacts on morbidity and mortality through study effects. In the current study, we take an alternative approach through which we use the case-control frequencies of the sickle cell trait (HbAS), the archetypal malaria resistance trait[8,9], as a proxy measure for the potential impact that an effective and specific malaria intervention might have on a wide range of common childhood conditions. Through this study we confirm that malaria has wider impacts on children's health than can not be attributed to the results of acute infections alone, a fact that should be taken into account when estimating the true burden of malaria on overall child health.

## Results

**Study population.** The study included 20,574 children < 14 years of age who were admitted to Kilifi County Hospital (KCH) from within the study area covered by the Kilifi Health and Demographic Surveillance System (KHDSS) during the 5-year period between 1st January 2000 and 31st December 2004. Of these potential cases, 18,864 (92%) were genotyped successfully for HbS and were included in the current analysis (Fig. 1). The baseline demographic, anthropometric and hematological characteristics of these children are summarized in Table 1. No systematic differences were seen with regard to these characteristics between genotyped and un-genotyped cases. Hospitalized HbAS children were younger, better nourished and less anemic than those with HbAA. Moreover, as expected, parasite densities were lower among those HbAS cases whose blood films were positive for *P. falciparum* malaria.

**Disease classification.** We classified the disease phenotypes of cases using both non-hierarchical and hierarchical approaches. The first approach allowed for multiple diagnoses within individual children while the latter classified each child with only a single phenotype on the basis of a hierarchy based on clinical and laboratory criteria (see Supplementary Table 1). The clinical phenotypes and outcome of hospital admission among non-hierarchically classified cases, stratified by HbAS genotype, are summarized in Table 2. Compared to those with HbAA, a substantially lower proportion of HbAS cases were admitted with a diagnosis of either malaria or severe malaria, and HbAS cases were approximately half as likely to be severely anemic or to receive a blood transfusion.

**Quantifying the non-specific contribution of malaria.** We then estimated the contribution of malaria to the risk of hospital admission, both overall and with a range of specific clinical syndromes, by using HbAS as a proxy-indicator for an effective malaria intervention. We did this by comparing the prevalence of HbAS among cases to that in a large group of community controls who were recruited from within the same geographic area as cases [$n$ = 4707: 4006 (85.1%) HbAA and 701 (14.9%) HbAS]. The results of these analyses are summarized as Odds Ratios (ORs) in Table 3. These analyses showed that HbAS was associated with significant protection against admission to hospital

both overall [adjusted OR (aOR) 0.62; 95% CI 0.55–0.69; $P$ < 0.0001, determined by logistic regression] and from admission with a wide range of specific diagnoses. While the effect size was greatest for conditions directly related to malaria, including malaria overall (aOR: 0.26; CI: 0.22–0.31; $P$ < 0.0001), strictly defined severe malaria (aOR: 0.14; CI: 0.09–0.22; $P$ < 0.0001), and severe anemia (aOR: 0.35; CI: 0.27–0.46; $P$ < 0.0001), HbAS was also associated with protection from admission with a range of conditions that could not easily have been directly attributable to malaria. For example, children with HbAS were less likely to be admitted to hospital with conditions associated with a negative malaria blood film (0.90; 0.83–0.98; $P$ = 0.011) or to be admitted during the first 28 days of life (0.79; 0.67–0.93; $P$ = 0.005). Furthermore, HbAS infants who were admitted during this period were significantly better nourished (weight-for-age Z score −1.59; −1.66, −1.52) than HbAA children (−1.69; −1.71, −1.67; $P$ = 0.006) (Table 1). Similarly, in comparison to those with HbAA, HbAS children were less likely to be admitted to hospital with bacteremia (0.69; 0.54–0.88; $P$ < 0.003) or to die during the course of their admission (0.69; 0.55–0.88; $P$ < 0.002). When analyzed on the basis of hierarchical definitions, the effect sizes remained similar for some conditions (including neonatal conditions and bacteremia) although they were reduced for others (Supplementary Tables 1 and 2). Using the same group of controls as a constant representation of the background prevalence of HbAS, we calculated the ORs for HbAS in cases versus controls in each year between 2000 and 2004. This is a period during which a significant decline in the prevalence of malaria was seen among children admitted to our hospital[10] (Fig. 2). While throughout this period the OR for HbAS remained constant among admitted children whose blood-films were positive for *P. falciparum* parasites, the ORs for HbAS among those admitted with a negative blood-film increased towards unity in parallel with the temporal decline in malaria.

## Discussion

The β$^s$ mutation of the β-globin (*HBB*) gene is the prime example of a balanced polymorphism in humans. Despite the negative effects of homozygosity (HbSS; sickle-cell anemia), a condition associated with high early mortality[11,12], the mutation has risen to high frequencies in many tropical areas because of the strong selective advantage afforded to heterozygotes (HbAS) against *P. falciparum* malaria. The protective effect of HbAS against the full range of malaria phenotypes is well documented, a recent meta-analysis having shown a consistent association with high levels of protection against both uncomplicated- and severe malaria in studies conducted throughout Africa[8]. Subsequently, it has been shown that the degree of malaria protection afforded by HbAS far exceeds that of any other polymorphism yet described[13,14]. As such, we believe that the malaria protective effect of HbAS is strongly analogous to that which might be seen with malaria-specific interventions, including an effective malaria vaccine. While it has been shown that under specific circumstances, HbAS can result in adverse consequences[15,16], it has never been associated with benefit against any other diseases in a non-malaria-endemic environment[16,17]. Specifically, while it has been shown that the risk of invasive bacterial infections is significantly reduced in HbAS children[10,18], this has only been observed in areas endemic for malaria[10]. As a consequence, it seems reasonable to assume that the association between HbAS and protection from admission to hospital with a wide range of clinical conditions that we saw in our study is explained by the indirect consequences of malaria on overall childhood health.

Although we would have liked to conduct a similar study in a non-malaria-endemic environment in order to confirm that the

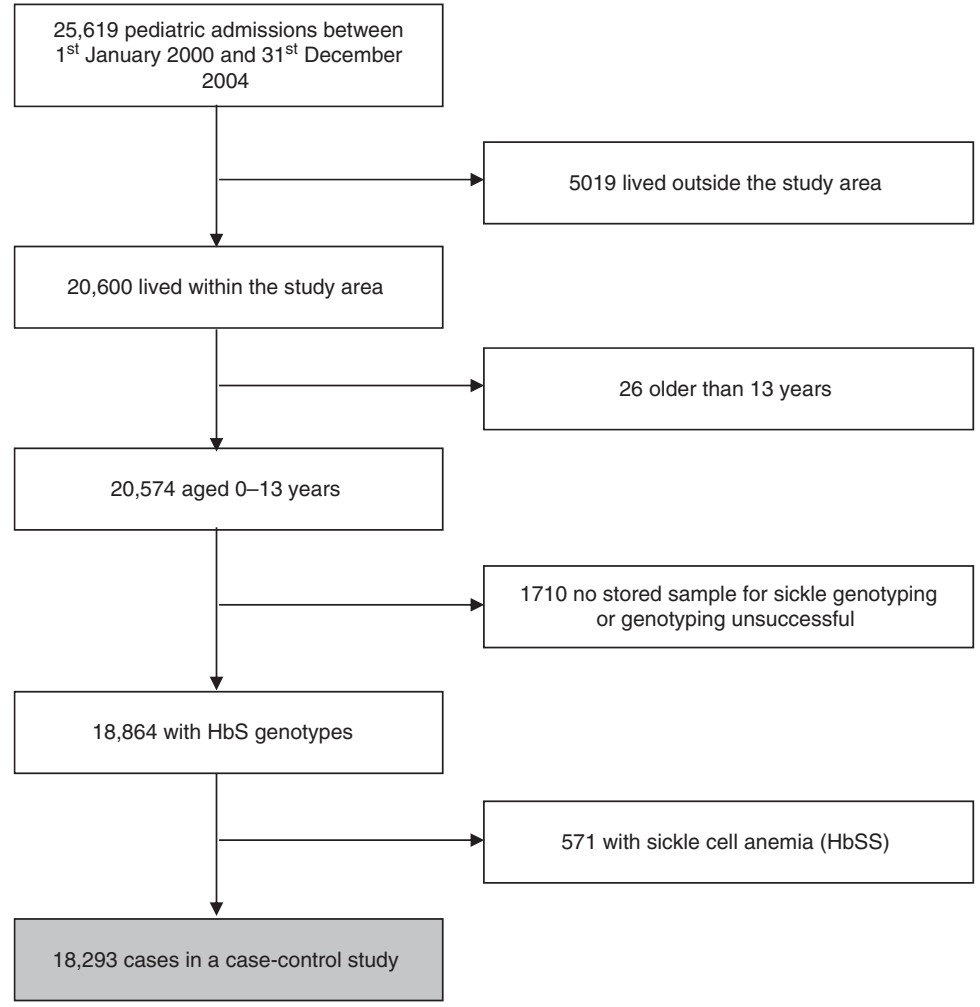

**Fig. 1** Profile showing the derivation of the case-patients included in the study. Figure shows the sample flow for patients contributing to the hospital admission data set

**Table 1 Clinical and laboratory characteristics of case-patients**

| Characteristic | HbAA | HbAS | P value |
|---|---|---|---|
| All admissions (n, %) | 16,502 (90.21) | 1791 (9.79) | n/a |
| Median age (IQR; months) | 17.5 (7.2–35.4) | 14.3 (5.2–31.8) | 0.0001 |
| Mean WAZ (95% CI) | −1.69 (−1.71, −1.67) | −1.59 (−1.66, −1.52) | 0.0075 |
| Mean HAZ (95% CI) | −1.33 (−1.36, −1.30) | −1.30 (−1.39, −1.22) | 0.613 |
| Mean parasite density (95% CI; parasites/mcl)[a] | 28,597 (27,015–30,270) | 10,002 (7686–13,015) | <0.0001 |
| Mean hemoglobin (95% CI g/L) | 91.4 (90.9–91.9) | 98.0 (96.6–99.3) | <0.0001 |
| Mean MCV (95% CI) | 74.1 (73.9–74.3) | 72.5 (71.8–73.2) | <0.0001 |
| Median WBC (IQR; ×10⁹/mcl) | 12.0 (8.7–16.7) | 12.9 (9.1–17.7) | <0.0001 |
| Median platelets (IQR; ×10⁶/L) | 273 (134–442) | 373 (227–528) | <0.0001 |

The data summarized reflect the clinical and laboratory features of hospital-admitted case-patients only, stratified by HbS genotype. P values estimated by use of Student's t test, χ², or Mann–Whitney tests as appropriate in comparison to HbAA children
*WAZ* weight-for-age Z-score, *HAZ* height-for-age Z-score
[a]Geometric mean densities among children with positive slides (7222 (43.8%) of HbAA and 384 (21.4%) of HbAS children)

wider impact of HbAS was attributable to malaria, we are not aware of any studies in which a large population of hospital-admitted children and representative community controls have been systematically tested for HbAS in a non-endemic region. Furthermore, it seems likely that in any such area the prevalence of HbAS would be too low to allow for meaningful interpretation. As an alternative therefore, we investigated the effect of changes in malaria transmission over time and observed that the

protection afforded by HbAS against a range of admission diagnoses correlated closely with malaria endemicity. This effectively eliminates pleiotropy or linkage disequilibrium as alternative explanations for our observations.

It has long been recognized that the impact of malaria on child mortality is greater than can be attributed to the direct consequences of malaria alone[7]. For example, effective malaria-control programs have frequently led to reductions in childhood mortality

**Table 2 Distribution of clinical syndromes and outcomes among case-children**

| Syndrome | HbAA (N = 16,502) | HbAS (N = 1791) |
|---|---|---|
| *Clinical syndromes* | | |
| Neonatal conditions | 1615 (9.8) | 224 (12.5) |
| Malaria | 5308 (32.2) | 252 (14.1) |
| Severe malaria | 838 (5.1) | 20 (1.1) |
| Severe pneumonia | 446 (2.7) | 54 (3.0) |
| Very severe pneumonia | 9760 (59.1) | 1073 (59.9) |
| Meningitis/encephalitis | 2779 (16.8) | 294 (16.2) |
| Severe malnutrition | 1428 (8.7) | 166 (9.3) |
| Gastroenteritis | 3038 (18.4) | 378 (21.1) |
| Transfused | 1543 (9.4) | 80 (4.5) |
| Other | 1069 (6.5) | 204 (11.4) |
| *Laboratory based syndromes* | | |
| Bacteremia | 854 (5.2) | 98 (5.5) |
| Malaria blood film positive | 7222 (43.8) | 384 (21.4) |
| Malaria blood film negative | 9280 (56.2) | 1407 (78.6) |
| Severe anemia | 1389 (8.6) | 80 (4.5) |
| *Outcome* | | |
| Median duration of admission (IQR; days) | 3 (2, 5) | 3 (2, 6) |
| Death (n; %) | 974 (5.9) | 113 (6.3) |

The data summarized reflect the clinical and laboratory features of hospital-admitted case-patients only, stratified by HbS genotype. Some children contribute data to more than one row. Definitions can be found in the text. Figures in parentheses denote the proportion of admissions with specific diagnoses within each genotypic group

that have been several-fold higher than those expected on the basis of prior estimates of malaria-specific mortality[7,19–21]. In one recent study, all-cause mortality was reduced by two-thirds within four years of introducing multiple intensive malaria control interventions[20]. This added value of malaria interventions almost certainly reflects the fact that malaria has important effects on child health more generally. For example, it is a recognized cause of malnutrition[22,23], a major determinant of all-cause mortality during childhood[5], and of invasive bacterial infections[10]. Nevertheless, proving this fact and estimating the magnitude of such effects can be difficult because few interventions are entirely specific to malaria. For example, those targeting the vector can also impact on other insect-transmitted diseases and some chemo-prophylactic agents such as sulfadoxine–pyrimethamine may also be active in preventing bacterial infections. Furthermore, it can also be difficult to control for study effects and secular trends.

In the current study, we have used the well-documented malaria-protective properties of HbAS as a proxy for estimating the indirect impact of malaria on a range of health outcomes in children admitted to hospital on the coast of Kenya. Because HbAS is strongly and specifically protective against malaria, we believe that our approach gives some indication of the likely benefits of malaria interventions, including an effective malaria-specific vaccine. HbAS was associated with a 38% reduction in hospital admissions overall, and while the greatest effect sizes were seen for conditions directly attributable to malaria, significant reductions were also seen for a number of outcomes that could not be readily attributed to its direct consequences. For example, admitted children with HbAS were significantly better nourished and were 10% less likely to be admitted with a negative malaria slide. Finally, it was also associated with a 21% reduction in the risk of admission in the neonatal period during which, for various reasons[24], clinical malaria is rare.

On the basis of the data from previous cohort studies, the protective effect of HbAS against uncomplicated clinical malaria (where malaria infections are not accompanied by signs of severity) is only around 30%[8], and is substantially lower against asymptomatic parasitemia[8]. Moreover, at the time of the study, malaria transmission within the study area was lower than in many other parts of sub-Saharan Africa. Consequently, although our study provides some indication of the wider impacts of malaria on child health, it is likely that both the true magnitude of these consequences within our study area and their impact in areas of higher transmission will be significantly greater. Similarly, while 31% protection against inpatient mortality provides some indication of the overall proportion of childhood deaths that are attributable to malaria, it is also likely that this would be significantly greater in areas of higher transmission.

One weakness of our study is our inability to provide diagnoses based on more detailed investigations. As in other hospitals in similar settings, diagnostic facilities at KCH are relatively limited. Most diagnoses, therefore, are based on clinical criteria alone and it can be difficult to differentiate with confidence between common conditions within the pediatric age range. For example, respiratory distress is a feature of both pneumonia, malaria and severe anemia and, as a consequence, it is likely that some of our categories include a significant degree of mis-classification. For this reason, we analyzed our study on the basis of both hierarchical and non-hierarchical definitions. While the former approach reduced the effect sizes for some, many associations remained significant including those for neonatal conditions, bacteremia and pneumonia.

Our observation that HbAS protected against admission during the neonatal period, and that HbAS neonates who were admitted were better nourished, is particularly interesting. Pregnant women are especially vulnerable to malaria, which can result in increased risks of low birth weight, prematurity and maternal and fetal mortality[25]. Moreover, malaria protection during pregnancy has been associated with a one fifth reduction in the risk of low birth weight[25]. However, these benefits are usually attributed to the prevention of malaria infection in mothers, while the observations regarding the benefits of HbAS in our current study relate exclusively to their children. We can envisage two potential explanations. The first relates to the fact that half of all HbAS children are born to HbAS mothers, and that our observation may simply reflect, therefore, the direct benefits of maternal malaria resistance. If this is the correct explanation then our observation must reflect a significant underestimate of the true benefits of maternal HbAS on pregnancy outcome[26]. Few studies have addressed this question directly and while none have found convincing evidence to support it[26–29], most have been too small to provide definitive conclusions. An alternative explanation is that our findings result from malaria resistance in the fetus. This is interesting because it is widely believed that the fetal consequences of malaria in pregnancy relate largely to infection within the placenta, and that direct infection of the fetus is relatively rare. Moreover, the switch from fetal hemoglobin (HbF: $\alpha_2\gamma_2$) to adult hemoglobin [either HbA ($\alpha_2\beta_2$) or HbS ($\alpha_2\beta^s_2$)] does not occur until after birth, making it hard to appreciate how HbS might offer protection prenatally. Nevertheless, this model may need to be revisited if this second explanation is correct.

Our study provides some insight into the impact of malaria on health systems in sub-Saharan Africa. Of particular note, despite the incomplete protection afforded by HbAS and the modest levels of malaria transmission at the time our study was conducted, malaria was probably responsible for at least 38% of admissions to KCH overall, and effective control would have resulted in a 65% reduction in the number of admissions with severe anemia. The latter is important for blood transfusion services in Africa, as evidenced by the fact that 68% fewer transfusions were administered to HbAS than to HbAA children.

At the same time as providing data on both the direct and indirect effects of malaria on child morbidity and mortality, our

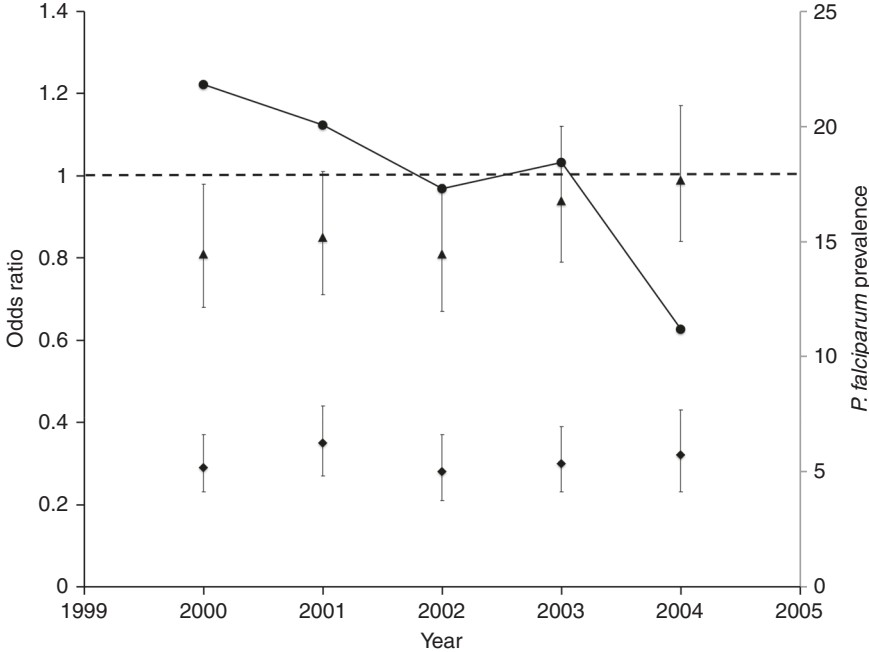

**Fig. 2** The odds ratios for HbAS in cases versus community controls, stratified by year of study. Triangles show the ORs for HbAS among patients admitted with a negative and diamonds for patients admitted with a positive blood slides for *P. falciparum* malaria parasites. Circles show the prevalence of *P. falciparum* parasite positivity among all admitted patients. The *x*-axis denotes each year of the study between 2000 and 2005. Horizontal dashed line denotes an OR of one: ORs below the line denote a protective effect of HbAS against the condition of interest. Error bars show 95% confidence intervals

| Table 3 The odds ratios for HbAS in case-patients versus community controls | | | | | |
|---|---|---|---|---|---|
| **Diagnosis** | **Proportion HbAS among cases (n/N, %)** | **OR (95% CI)** | **P value** | **aOR (95% CI)** | **P value** |
| *Clinical syndromes* | | | | | |
| All cause hospital admission | 1791/18,293 (9.8) | 0.62 (0.56–0.68) | < 0.0001 | 0.62 (0.55–0.69) | < 0.0001 |
| Neonatal conditions | 224/1839 (12.2) | 0.79 (0.67–0.93) | 0.005 | 0.79 (0.67–0.93) | 0.005 |
| Malaria | 252/5560 (4.5) | 0.27 (0.23–0.32) | < 0.0001 | 0.26 (0.22–0.31) | < 0.0001 |
| Severe malaria | 20/858 (2.3) | 0.14 (0.09–0.21) | < 0.0001 | 0.14 (0.09–0.22) | < 0.0001 |
| Severe pneumonia | 54/500 (10.8) | 0.69 (0.52–0.93) | 0.014 | 0.70 (0.51–0.96) | 0.026 |
| Very severe pneumonia | 1073/10,833 (9.9) | 0.63 (0.57–0.70) | < 0.0001 | 0.64 (0.56–0.72) | < 0.0001 |
| Meningitis/encephalitis | 294/3073 (9.6) | 0.60 (0.52–0.70) | < 0.0001 | 0.60 (0.50–0.72) | < 0.0001 |
| Severe malnutrition | 166/1594 (10.4) | 0.66 (0.56–0.80) | < 0.0001 | 0.67 (0.55–0.83) | < 0.0001 |
| Gastroenteritis | 378/3416 (11.1) | 0.71 (0.62–0.81) | < 0.0001 | 0.72 (0.61–0.86) | < 0.0001 |
| Other | 204/1273 (16.0) | 1.09 (0.92–1.29) | 0.32 | 1.08 (0.90–1.29) | 0.41 |
| *Laboratory features and outcomes* | | | | | |
| Bacteremia | 98/952 (10.3) | 0.66 (0.52–0.82) | < 0.0001 | 0.69 (0.54–0.88) | 0.003 |
| Malaria blood film positive | 384/7606 (5.0) | 0.30 (0.27–0.35) | < 0.0001 | 0.30 (0.26–0.35) | < 0.0001 |
| Malaria blood film negative | 1407/10,687 (13.2) | 0.87 (0.79–0.96) | 0.004 | 0.90 (0.83–0.98) | 0.011 |
| Severe anemia | 80/1469 (5.4) | 0.33 (0.26–0.42) | < 0.0001 | 0.35 (0.27–0.46) | < 0.0001 |
| Transfused | 80/1623 (4.9) | 0.30 (0.23–0.38) | < 0.0001 | 0.32 (0.25–0.41) | < 0.0001 |
| Died | 113/1087 (10.4) | 0.66 (0.54–0.82) | < 0.0001 | 0.69 (0.55–0.88) | 0.002 |

OR: crude odds ratios for HbAS were derived through comparison of the genotype frequencies for HbAA and HbAS in cases versus community controls using logistic regression. Those for cases are shown in the second column while the frequency of HbAS among controls was 701/4707 (14.9%). aOR is the odds ratios adjusted for age groups defined as 0–3 years, 4–7 years and 8–12 years. The genotype frequencies for the control group, stratified by age category, are described in more detail under methods. *P* values were estimated using logistic regression

study illustrates the wider selective advantage of HbAS in malaria-endemic environments. It has been suggested that the time depth for selection of the β[s] mutation is too short to be fully explained by malaria alone[30]. Our data on the added value of HbAS in malaria-endemic environments help to reconcile this previous observation.

In summary, through a large case-control study of childhood diseases conducted in a malaria-endemic area on the coast of Kenya, we present new evidence to quantify the extended impact of malaria against childhood morbidity and mortality. Our study supports the conclusion that effective interventions will result in health benefits that could considerably exceed those directly attributable to malaria alone[2,7].

## Methods

**Study design.** We investigated the overall impact of malaria on child health through a case-control approach in which we used HbAS as a proxy indicator for malaria exposure. A system of routine clinical surveillance has been operating at Kilifi County Hospital (KCH) since 1989 through which all children are assessed at both admission and discharge using a standard computerized proforma. In addition, a battery of routine tests are conducted on all children at admission, including a full blood count, a malaria blood-film and a blood culture[31]. Data and samples

are systematically archived which allows for the conduct of retrospective studies such as this one. For the current study, we defined cases as children aged <14 years who were admitted from the area served by the Kilifi Health and Demographic Surveillance System (KHDSS)[32] onto the wards of KCH between 1st January 2000 and 31st December 2004, a period during which the incidence of uncomplicated *P. falciparum* malaria among children within this region was between 1 and 3 episodes/person/year[33], allowing us to study the research hypothesis in question. Data derived from this surveillance system allowed us to further classify cases into specific disease categories on the basis of both clinical and laboratory criteria. Clinical syndromes were defined as follows. Neonatal conditions were defined as admission to hospital within the first 28 days of life; malaria as a fever in the presence of *P. falciparum* parasitemia at any density in children <1 year old or at a density of >2500 parasites/μl in older children; severe malaria as malaria in association with the specific complications of prostration and/or coma and/or respiratory distress and/or a Hb of <50 g/L. Severe and very severe pneumonia were defined using standard methods[34] while meningitis/encephalitis was defined by the presence of neck stiffness, a bulging fontanelle, prostration or coma (defined as a Blantyre Coma Score of <5). Severe malnutrition was defined on the basis of a mid-upper-arm circumference of ≤7.5 cm in children < 6 months or of ≤11.5 in children ≥6 months of age. Finally, gastroenteritis was defined as diarrhea ( ≥ 3 loose watery stools/day) with or without vomiting (≥3 episodes/day). For the purposes of our case-control analyses, we assembled a control group that was representative of cases in terms of their area of residence and age. Given that this was a retrospective study, we did not have access to samples from children precisely matched in terms of time of sampling, age and other variables. Instead, we constructed a control panel consisting of unselected children who were recruited into a range of epidemiological studies that were conducted throughout the KHDSS study area between September 1998 and November 2005, for whom data on HbS phenotype (HbAA or HbAS) had been recorded for the purposes of those studies, and who were 0–13 years of age at the time of sampling[33,35–37]. The resulting control group included a total of 4707 children with the following age structure and HbS frequencies: 0–3 years 1364 (29.0%) [1163 (85.3%) HbAA and 201 (14.7%) HbAS]; 4–7 years 2177 (46.3%) [1859 (85.4%) HbAA and 318 (14.6%) HbAS]; 8–13 years 1166 (24.8%) [984 (84.4%) HbAA and 182 (15.6%) HbAS].

**Ethics**. Informed written consent was obtained from all study participants, their parents or guardians. Ethical permission for this study was granted by the KEMRI/ National Ethics Research Committee in Nairobi, Kenya.

**Laboratory procedures**. Hematological, biochemical and malaria parasite data were derived by standard methods[38] while blood cultures were processed in BACTEC Peds Plus bottles using a BACTEC 9050 automated blood-culture instrument (Becton Dickinson, UK). Positive samples were sub-cultured on standard media by routine microbiological techniques[31]. Quality assurance for all laboratory tests was provided by the UK National External Quality Assessment Service (www.ukneqas.org.uk). Within cases, we retrospectively tested for HbAS by PCR[39] using DNA extracted from fresh or frozen samples of whole blood using proprietary methods [ABI PRISM (Applied Biosystems, California, USA) or Qiagen DNA Blood Mini Kit (Qiagen, West Sussex, United Kingdom)]. Throughout the study, therefore, admitting clinicians were unaware of the HbS status of cases. For controls, we tested for HbAS, within 7 days of recruitment, using fresh blood samples collected into EDTA by alkaline electrophoresis on cellulose acetate gels (Helena Titan™III, Helena Biosciences, Gateshead, UK) using standard methods.

**Statistical analysis**. Because of its association with major health consequences, for the purpose of the current analysis we excluded children with sickle cell anemia (HbSS) from both the case and control groups. We compared the clinical, laboratory and demographic characteristics of HbAA vs. HbAS children who were admitted to hospital during the study period (cases) using parametric or non-parametric tests as appropriate, while proportions were compared using the $\chi^2$ test. To estimate the impact of malaria on admission to hospital both overall and with a range of specific conditions, we calculated the odds ratios (ORs) for HbAS in cases versus community controls by logistic regression, both with and without adjustment for age, categorized as 0–3 years, 4–7 years and 7–14 years. All analyses were conducted using Stata v11.2 (Stata Corp, Timberlake).

**Code availability**. The code associated with the statistical analysis of the primary data, written in Stata v11.2, have been deposited on the data repository for the KEMRI/Wellcome Trust Research Programme in Kilifi and are available, along with appropriately anonymized data, by application through MMunene@kemri-wellcome.org.

**Reporting Summary**. Further information on experimental design is available in the Nature Research Reporting Summary linked to this article.

## Data availability
Requests for access to appropriately anonymized data from this study can be made by application to the data governance committee at the KEMRI/Wellcome Trust Research

Progamme through the following e-mail address: <MMunene@kemri-wellcome.org>. The authors declare that all other data supporting the findings of this study are available within the article and its Supplementary Information files, or are available from the authors upon request.

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

## Acknowledgements
We thank Johnstone Makale, Emily Nyatichi, and Metrine Tendwa for laboratory support, Kevin Marsh for his support in developing the surveillance platform, all members of staff at Kilifi County Hospital and the KEMRI–Wellcome Trust Research Programme, Kilifi, who helped with data and sample collection, and the study participants and their parents for participating in this study. T.N.W. and J.A.G.S. are funded through Senior Research Fellowships from the Wellcome Trust (091758, 202800, 098532), who also provided core support to the KEMRI–Wellcome Trust Research Programme in Kilifi (203077). This paper is published with permission from the Director of the Kenya Medical Research Institute.

## Author contributions
T.N.W., S.U., J.A.G.S. and K.M. designed the study and conducted the literature review. A.W.M. and K.O.A. assisted with sample preparation and analysis. M.S., N.M., N.P. and B.T. assisted with the collection and interpretation of clinical data. T.N.W., C.M.N., G. N., G.M.C. and G.N. analyzed the data and all authors helped to interpret the data. T.N. W. and S.U. wrote the first draft of the report and all authors contributed to editing the final version.

## Additional information

**Competing interests:** The authors declare no competing interests.

