## [Peer Review File · Nature Communications]

Reviewers' Comments:

Reviewer #1:

Remarks to the Author:

I previously reviewed this manuscript for a different journal. It's disappointing to see some minor but clear errors that have not been corrected - for example, on page 5, the authors report "adjusted OR (aOR) 0.65; 95% Confidence Interval (CI) 0.60,0.65; $P < 0.0001$ ". Having spent time making suggestions to improve the manuscript, it's frustrating for these suggestions to be ignored. As an aside, I am a statistician and am unaware of previous research in the field of malaria - I can comment on the validity and clarity of analyses, but not on the potential relevance or impact of this work.

This is an interesting paper designed to investigate the broad consequences of malaria - not just clinical malaria, but also subclinical malaria, and not just the narrow consequences of malaria, but all the negative health consequences that accompany malaria. This is achieved not by comparing individuals with malaria to those without malaria, but by comparing those with the sickle-cell mutation to those without the sickle-cell mutation - a mutation that confers a high degree of protection for malaria.

Major points:

1. The authors frame this as a Mendelian randomization investigation. I feel that this framing is unhelpful, partly because the terminology of Mendelian randomization is not universally understood, and partly because the risk factor is not modifiable - the initial definitions of Mendelian randomization pre-suppose that the risk factor is a modifiable one. I think a better framing of the investigation would be one using the sickle-cell mutation as a naturally-occurring proxy measure of increased natural resistance to malaria -- without requiring the authors to appeal to the framework of MR. It is a natural experiment with some similarity to MR, but I don't think that claiming this is an MR investigation will be helpful.

2. About the suitability of the genetic variant as a proxy for malaria resistance - the authors claim that the variant is "strongly and specifically protective against malaria" (p9) - similar sentiments are found in other places. Is this a fair statement? My naive understanding was that sickle trait had broader phenotypic consequences - but perhaps these are mediated via malaria (ie part of the same pathway). To what extent is comparing sickle-null individuals with sickle-trait individuals an analogous comparison to evaluating the effect of malaria-reducing interventions? In particular, to what extent is the sickle-trait mutation a good proxy for malaria vaccination?

3. Can I please clarify the tables and figures? Figure 1 - is this the cases only? (May be good to show where the controls come from too.) Table 1 and Table 2 - this is just within the cases? And Table 3 - I don't fully understand the comparison here. Is this HbAS children only? Or is this HbAA vs HbAS? The title suggests HbAS only, but in that case I don't get what comparison is being made. Is this in cases and controls? (If not, then in which analysis have you used the controls?) And Figure 2 - what does the odds ratio represent? The increased odds of being admitted to hospital for HbAS versus HbAA? So an odds ratio of 1 means that an HbAS individual is as likely to be hospitalized as an HbAA individual? And a negative odds ratio means that an HbAS individual is less likely to be hospitalized than an HbAA individual? Have I interpreted correctly?

4. Is there a way of investigating mediation here? For example, is the effect of malaria resistance on reduced infection mediated via lower anaemia risk? (I'm not sure if this is possible given the data available, but it seems like an interesting question.)

Minor points:

p6: "adjusted OR (aOR) 0.65; 95% Confidence Interval (CI) 0.60,0.65; P<0.0001" - the estimate shouldn't be the same as the upper CI limit - please check this.
p7: WAZ - please spell out/define.

Stephen Burgess

Reviewer #2:

Remarks to the Author:

Williams and colleagues report a case-control study to quantify the effects of sickle-trait hemoglobin in a malaria-endemic area on non-malaria health outcomes. The rationale for doing so is that it is generally hypothesized/believed that malaria has many ill effects on non-specific health outcomes that are difficult to directly measure; herein, the authors use sickle-trait as an upstream instrumental variable with the premise that any association of sickle-trait with outcomes exerts its effect through malaria, because sickle-trait has fairly silent clinical morbidity. This is a sound premise, and a clever approach. The results of such a study would be a sound contribution to highlighting the insidious effects of malaria on general childhood health as well as helping to quantify the contribution of malaria to these poor outcomes.

A further benefit of this approach is that it leverages the significant assets of the Kilifi team, who have nearly unmatched facility- and community-based data from a (previously) malaria-endemic region with which to conduct a study such as this.

My major concern with this study is the appropriateness of the control group, about which very little is said. This is critical of course for a case-control study, and I'm not convinced that the controls - despite being from the same area as the cases - are appropriate. The reason that this is critical is that the major takeaways from the report are derived from a comparison of cases (children with AS admitted to hospital 2000-2004) and controls (children with AS enrolled into a community cohort 2006-2010); these major takeaways are presented in Table 3, in which sickle-trait is associated with reductions in a wide variety of clinical syndromes that are not typically associated with malaria, the interpretation of which is that malaria is a risk factor for these other outcomes (severe pneumonia, severe malnutrition, gastroenteritis, etc).

But it is also possible that in comparing trait children under 14y from 2000-2004 with trait children under 5y from 2006-2010, that there are secular trends in these other outcomes, or simply bias introduced by age differences between the groups. What is not presented is an analogous Table 3 for non-trait children; if such ORs demonstrated that there was no differences among AA children on these outcomes between cases and controls, but there was for AS children, then that would be convincing evidence that trait is the meaningful difference and not the many potential biases that are inherent in comparing such different cases and controls.

Similarly, there may have been differences in access to health care between the two cohorts, specifically better access to primary care in the control cohort, or more active preventive care, which would be expected to reduce the likelihood of admission for these various severe problems.

Therefore, much more detail needs to be provided to make it clear that the controls are appropriate for the case series, and more argument that the differences in risk of outcomes for the trait kids are not the result of systematic but unmeasured or unmeasurable differences between groups in what is

essentially an ecological study between two time periods.

Other comments:

1. For transparency, all Tables I think should have n/N for the various conditions, this will also improve rigor of reporting and clarity.
2. It is surprising to me that, with only a $\sim 30\%$ protection from uncomplicated malaria attributable to AS (and no protection from parasitization), the authors report in Table 3 even greater degrees of protection from these other non-malaria outcomes, that would be most likely to have been the result of recurrent uncomplicated malaria or parasitization. The authors' speculation on these inconsistent degrees of protection would be welcome.
3. Lines 200-203, the OR for uncomplicated malaria is 0.70, which indicates a 30% reduction in risk with AS.
4. The analysis represented in Fig 2 and its significance is honestly lost on me.

Reviewer #1

1) The authors frame this as a Mendelian randomization investigation. I feel that this framing

is unhelpful, partly because the terminology of Mendelian randomization is not universally understood, and partly because the risk factor is not modifiable - the initial definitions of Mendelian randomization pre-suppose that the risk factor is a modifiable one. I think a better framing of the investigation would be one using the sickle-cell mutation as a naturally-occurring proxy measure of increased natural resistance to malaria -- without requiring the authors to appeal to the framework of MR. It is a natural experiment with some similarity to MR, but I don't think that claiming this is an MR investigation will be helpful.

Coincidentally, Dr Burgess had previously received our paper for review when under consideration by another journal. Interestingly, we received 6 widely differing reviews from that journal, not all of which could easily be reconciled with the views expressed by Dr Burgess regarding this point. Furthermore, we did consider his comments in the context of our previous experiences of publishing papers in which we have used the same concept (eg Scott *et al*, Lancet 2011, PMC3192903), in which editors and reviewers have often encouraged us to use the MR framework. Nevertheless, we do recognize that Dr Burgess is an expert in this area, accept that the MR concept is not particularly intuitive, and also agree that the message is more easily communicated through the alternative framework that he has suggested. On that basis, we have re-drafted the paper under this alternative framework on the assumption that the editors and other reviewers would be happy with this alternative approach. We have rephrased our manuscript throughout, from the abstract onwards, to reflect this change of framework, in which we now refer to the sickle-cell mutation as a naturally-occurring proxy measure of increased natural resistance to malaria.

2) About the suitability of the genetic variant as a proxy for malaria resistance - the authors claim that the variant is "strongly and specifically protective against malaria" (p9) - similar sentiments are found in other places. Is this a fair statement? My naive understanding was that sickle trait had broader phenotypic consequences - but perhaps these are mediated via malaria (ie part of the same pathway). To what extent is comparing sickle-null individuals

with sickle-trait individuals an analogous comparison to evaluating the effect of malaria-reducing interventions? In particular, to what extent is the sickle-trait mutation a good proxy for malaria vaccination?

As presented in the first paragraph of the discussion, the protective effect of HbAS against the full range of clinical *P. falciparum* malaria phenotypes is among the highest of any infectious disease associated gene ever described. This is well captured in the recent meta-analysis by Taylor and colleagues which we have cited in our paper (Reference (8)). In comparison to HbAA subjects, those with HbAS have only approximately 30% of the risk of uncomplicated malaria and 10% of the risk of severe and complicated malaria (case-control ORs of ~0.3 and ~0.1 respectively). The only malaria vaccine that is at an advanced stage in its development provides, at most, 50% protection against uncomplicated malaria for a limited period of time (up to one year). We therefore believe that the malaria-protective effects of HbAS are strongly analogous to those of malaria reducing interventions and considerably more instructive than the current malaria vaccine. Although we think that this is already captured sufficiently in the first paragraph of the discussion, we have now added an extra sentence to that paragraph to reinforce this message which reads as follows: “.....As such, we believe that the malaria protective effect of HbAS is strongly analogous to that which might be seen with malaria-specific interventions, including an effective malaria vaccine..”

3a) Can I please clarify the tables and figures? Figure 1 - is this the cases only? (May be good to show where the controls come from too.)

We apologise for our lack of clarity. The reviewer is correct - these are just the cases. We have altered the title of this figure to make this more clear which now reads “Profile showing the derivation of the case-patients included in the study”. In response to the concerns of reviewer 2, we have now used a different set of controls which we also now describe in greater detail in the methods section of the text (see response to reviewer 2 below).

3b) Table 1 and Table 2 - this is just within the cases?

Yes – Tables 1 and 2 summarise the data for cases only. Although this is already stated in the titles of these tables we have altered them for greater clarity and added a sentence to the footnotes to each table that emphasizes this point. These footnotes now read: “The data summarized reflect the clinical and laboratory features of hospital-admitted case-patients only, stratified by HbS genotype.”

3c) And Table 3 - I don't fully understand the comparison here. Is this HbAS children only? Or is this HbAA vs HbAS? The title suggests HbAS only, but in that case I don't get what comparison is being made. Is this in cases and controls? (If not, then in which analysis have you used the controls?)

We apologize for our lack of clarity. The reviewer has interpreted the data correctly: the table summarises the ORs for HbAS in cases versus controls. We have altered the title to make this more clear which now reads as follows: “The odds ratios for HbAS in cases admitted to hospital versus community controls.” We have also strengthened the footnote to this table which now reads “OR: crude odds ratios for HbAS were derived through comparison of the genotype frequencies for HbAA and HbAS in cases versus community controls using logistic regression; aOR: odds ratios adjusted for age groups defined as 0-3 years, 4-7 years and 8-12 years.”

3d) And Figure 2 - what does the odds ratio represent? The increased odds of being admitted to hospital for HbAS versus HbAA? So an odds ratio of 1 means that an HbAS individual is as likely to be hospitalized as an HbAA individual? And a negative odds ratio means that an HbAS individual is less likely to be hospitalized than an HbAA individual? Have I interpreted correctly?

Yes – this is the correct interpretation. We have added a footnote to the figure to make this clearer. We have also considerably expanded the footnote to explain how the analyses were conducted and have also included the proportions of cases and controls with HbAS which addresses a comment also made by reviewer 2.

4) Is there a way of investigating mediation here? For example, is the effect of malaria resistance on reduced infection mediated via lower anaemia risk? (I'm not sure if this is possible given the data available, but it seems like an interesting question.)

We agree that this would be a very interesting point; however, we don't believe there is a way to investigate mechanisms by use of the data available. We have not made any changes in the light of this question, although we would be happy to elaborate further in the discussion if this would be considered helpful.

Minor points:

p6: "adjusted OR (aOR) 0.65; 95% Confidence Interval (CI) 0.60,0.65; P<0.0001" - the estimate shouldn't be the same as the upper CI limit - please check this.

We apologize. This has now been corrected in the revised analysis.

p7: WAZ - please spell out/define.

We have spelled out in full throughout.

Reviewer #2

1) My major concern with this study is the appropriateness of the control group, about which very little is said. This is critical of course for a case-control study, and I'm not convinced that the controls - despite being from the same area as the cases - are appropriate. The reason that this is critical is that the major takeaways from the report are derived from a comparison of cases (children with AS admitted to hospital 2000-2004) and controls (children with AS enrolled into a community cohort 2006-2010); these major takeaways are presented in Table 3, in which sickle- trait is associated with reductions in a wide variety of clinical syndromes that are not typically associated with malaria, the interpretation of which is that malaria is a risk factor for these other outcomes (severe pneumonia, severe malnutrition, gastroenteritis, etc). But it is also possible that in comparing trait children under 14y from 2000-2004 with trait children under 5y from 2006-2010, that there are secular trends in these other outcomes, or simply bias introduced by age differences between the groups. What is not presented is an analogous Table 3 for non-trait children; if such ORs demonstrated that there was no differences among AA children on these outcomes between cases and controls, but there was for AS children, then that would be convincing evidence that trait is the meaningful difference and not the many potential biases that are inherent in comparing such different cases and controls. Similarly, there may have been differences in access to health care between the two cohorts, specifically better access to primary care in the control cohort, or more active preventive care, which would be expected to reduce the likelihood of admission for these various severe problems. Therefore, much more detail needs to be provided to make it clear that the controls are appropriate for the case series, and more argument that the differences in risk of outcomes for the trait kids are not the result of systematic but unmeasured or unmeasurable differences between groups in what is essentially an ecological study between two time periods.

From the above comments it does not appear that Reviewer 2 has completely understood the nature of our analysis. The controls in this analysis are used simply as a group through which to compare the frequency of HbAS among cases and controls, as described by your statistical reviewer above. We have done this, as describe in the paper and in response to reviewer 1, as a proxy indicator for the role of malaria. The analogous table for HbAA would therefore simply be the reciprocal of table 3. From some of the comments made by reviewer 2 perhaps the reviewer is thinking we had conducted a cohort analysis? Although we have explained in the paper why we believe our controls are appropriate for this analysis, we do acknowledge that the use of a control group that has both a different age structure and that was recruited at a different point of time is an unnecessary distraction, and we also acknowledge that it does introduce a degree of bias as a result

of the differential survival of HbAA and HbAS children. In reality, the bias that this might introduce – failing to capture the rising proportion of HbAS with age among the control population - would actually make it more difficult to identify protective associations. Nevertheless, in the light of this reviewer's comments, and similar comments made by other reviewers when previously submitted elsewhere, we have substituted our controls with an alternative set, derived from children 0-13 years of age who were recruited by random sampling throughout the Kilifi Epi-DSS study area into a number of different studies conducted between September 1998 and November 2005. We describe how we assembled this set of control children in much more detail in the methods section of the revised paper. This alternative set of controls are now more contemporaneous with the cases and also enable us to adjust for the minor increase in the frequency of HbAS within the population that occurs with age - a rise from 14.7% in the youngest children to 15.6% in the oldest children. We have now completely revised the case-control analysis presented in the text and in Table 3 and Figure 2 of our paper to reflect this alternative set of controls. In practice, the new analysis gives very similar outputs and makes no material difference to our conclusions. We have also amended the text within the methods and results sections to make the nature of our analysis clearer.

2) For transparency, all Tables I think should have n/N for the various conditions, this will also improve rigor of reporting and clarity.

We are unclear here about what the reviewer is asking for. In Table 1, the total number of children admitted to hospital during the study period who are included in the analysis was 18,864 (as stated in the results section) and the numbers stratified by HbS genotype are stated in the first line under the title "all admissions". The data summarized in the subsequent rows are for all children in those columns, with the exception of the malaria parasitaemia data that only applies to those who were parasite positive. We have added the numbers for parasite positives in the footnote of this table. In Table 2, the n's (numerators) for each individual condition by HbS genotype are shown in the cells while the overall Ns (denominators) for each genotypic group are shown in the header (16,502 for HbAA and 1,791 for HbAS). Finally, the n/Ns for Table 3 can be concluded from Table 2 (which shows the n's for cases) and from the genotype frequencies in controls. We have now added a note in the footnote to table 3 to make this more clear, including a clear description of what we have done in the analysis and a summary of the HbAS frequencies among children in each disease category.

3) It is surprising to me that, with only a ~30% protection from uncomplicated malaria attributable to AS (ad no protection from parasitization), the authors report in Table 3 even greater degrees of protection from these other non-malaria outcomes, that would be most likely to have been the result of recurrent uncomplicated malaria or parasitization. The authors' speculation on these inconsistent degrees of protection would be welcome.

We are confused by this comment and are not sure where the reviewer has derived these figures. The data in Table 3 show that HbAS was associated with highly significant OR of 0.14 and 0.26 for severe and uncomplicated malaria respectively and of 0.30 for malaria parasitaemia. These suggest protection of 70% or more for all forms of clinical malaria. We would be grateful for clarification.

4) Lines 200-203, the OR for uncomplicated malaria is 0.70, which indicates a 30% reduction in risk with AS.

Our apologies. We have changed the figure to read 30% instead of 70%.

5) The analysis represented in Fig 2 and its significance is honestly lost on me.

We have expanded and altered the explanations about how and why we conducted this analysis and how we interpret the results within the results, discussion and methods sections and in a new foot note to Figure 2. We hope this clarifies this point.

Reviewers' Comments:

Reviewer #1:

Remarks to the Author:

Thank you for the opportunity to review this paper again. The primary result of the paper is that, in a comparison of hospital cases and community controls, an individual with a HbAS genotype is more likely to be a community control than to be hospital case (and conversely, an individual with a HbAA genotype is more likely to be a hospital case). This is the straightforward finding of the statistical analysis with no inference or interpretation. The association is stronger for malaria and severe malaria, but also present for other conditions that are not (and here I am trusting the authors) sequelae of malaria. I'm slightly suspicious of the authors' claim that these conditions (eg severe malnutrition) are unrelated to malaria (in the authors' words "not readily attributable to malaria") as I believe these conditions are comorbid with malaria. But I am not the right person to judge this claim. The authors then infer: i) that a malaria vaccination could result in individuals in the population being protected from malaria so that they are similar to the HbAS genotype individuals, and ii) that the effects of malaria have been underestimated in previous work.

While I appreciate the authors' work, I have some lingering concerns:

1) Returning to my previous point 2, my understanding was that individuals with heterozygous HbAS (sickle-cell trait) are phenotypically different from non-sickle cell carriers in ways that do not relate to malaria - they get shortness of breath, and there is some impairment in terms of oxygen transport around the body. Is this correct? This is a memory from my school-age biology classes, so I may well be incorrect here. The issue is that you are comparing HbAA and HbAS and making inferences about malaria vaccination - the question is whether HbAS individuals differ in other ways than malaria susceptibility from HbAA individuals. As if they do, then you cannot attribute the entirety of the genetic association to malaria vs non-malaria.

2) About underestimation, genetic variants are typically associated with larger phenotypic changes than clinical interventions. There are several reasons for this - genetic changes are lifelong, and they are not subject to attenuation of effect due to failure to administer the intervention correctly, non-compliance, and so on. So I wonder how much the genetic analysis suggests that previous estimates of the effectiveness of malaria vaccination have been underestimated, and how much this is simply telling us the effectiveness of an impossibly optimistic intervention - one administered at conception with no non-compliance. The impact of this sort of intervention is valuable to know about, but we can't expect similar impact in practice.

Otherwise, the paper is interesting, and much clearer after review - particularly with the more relevant control group. My only concern is that the reviewers (including myself) have concentrated on the epidemiological aspects of the paper - have the clinical aspects been adequately reviewed?

Stephen Burgess

Reviewer #2:

Remarks to the Author:

This remains a potentially important paper. And several concerns have been partially addressed. I personally found the Mendelian randomization framework useful, but the other Reviewer appears to have objected.

1. Re: selection of appropriate controls, the authors appear to have mitigated this major risk with the application of more germane controls. However, the description of them continues to confuse. Lines 304-307 indicate that "children aged 3-12 months of age who were born within the same study area as cases between August 2006 and September 2010... were used as controls," but then lines 311-315 state that "we constructed a control panel consisting of unselected children... between September 1998 and November 2005..." Which is it? Were there two control panels?

2. Table 3 can still be clarified by n/N that were used to compute at least the unadjusted ORs. I note that reviewer 1 also commented on the lack of clarity in Table 3, which I don't think has been improved substantially by the footnote. This transparency will help readers understand what is really the meat of the analysis, and enable meta-analyses by future groups, if people want to do that sort of thing. Even if it is derivable from elsewhere in the manuscript, I don't see a barrier to including them here.

Minor

1. Lines 122-124. In this paragraph you have moved on to describing Table 3 results, but then in these lines you revert back to the prior analysis of case children only, which are results that you presented in a prior paragraph. Why go back?

2. Lines 131-134. This may be a question of journal policy, but I don't think it is good form to describe statistically insignificant findings as "marginal." It would be more appropriate to describe them as "decreases in risk that were not statistically significant" (irrespective of clinical significance).

Reviewer #1

1) Returning to my previous point 2, my understanding was that individuals with heterozygous HbAS (sickle-cell trait) are phenotypically different from non-sickle cell carriers in ways that do not relate to malaria - they get shortness of breath, and there is some impairment in terms of oxygen transport around the body. Is this correct? This is a memory from my school-age biology classes, so I may well be incorrect here. The issue is that you are comparing HbAA and HbAS and making inferences about malaria vaccination - the question is whether HbAS individuals differ in other ways than malaria susceptibility from HbAA individuals. As if they do, then you cannot attribute the entirety of the genetic association to malaria vs non-malaria.

We can reassure the reviewer that to all intents and purposes, subjects with sickle cell trait are phenotypically indistinguishable from those without sickle cell trait. Subjects with sickle cell trait do, clearly, differ from non-trait individuals in one important regard: that their red blood cells contain, on average, around 40% HbS and 60% HbA in comparison to those of non-trait individuals which contain close to 100% HbA. However, under normal physiological circumstances they most certainly do not get shortness of breath nor do they have any demonstrable impairments in terms of tissue oxygen delivery. From a physiological perspective, those with sickle cell trait are every bit as fit as those without and the condition is common among elite black athletes and military personnel. Perhaps the reviewer is remembering the rare complications that can occur under circumstances of profound and prolonged hypoxaemia such as non-pressurized high-altitude flight, extreme exercise or hypoxaemic general anaesthesia? While we appreciate the author's comment, we are at a loss to know how to edit our manuscript in a way that will capture this "lingering concern". We would be happy to consider suggestions.

2) About underestimation, genetic variants are typically associated with larger phenotypic changes than clinical interventions. There are several reasons for this - genetic changes are lifelong, and they are not subject to attenuation of effect due to failure to administer the intervention correctly, non-compliance, and so on. So I wonder how much the genetic analysis suggests that previous estimates of the effectiveness of malaria vaccination have been underestimated, and how much this is simply telling us the effectiveness of an impossibly optimistic intervention - one administered at conception with no non-compliance. The impact of this sort of intervention is valuable to know about, but we can't expect similar impact in practice.

While the reviewer's statement is certainly true for some conditions it is not true for infectious diseases. In our field of genetic epidemiology there are no examples of genetic variants that have larger phenotypic effects on disease risk than effective interventions. Take for example the case of common bacterial infections such as *Haemophilus influenzae* or *Streptococcus pneumoniae* – the newest vaccines confer between 80 and 95% protection against invasive forms of these conditions. Similarly, a single course of tetanus, polio or yellow fever vaccines essentially confer complete life-long cover from these diseases. Nevertheless, with the exception of sickle cell trait, no genetic factors have ever been described that alter the risk of any of these diseases by more than a few percentage points. An effective malaria vaccine should do just as these other vaccines do – to provide long-term high-level protection to those who are vaccinated (we are not in the current study tackling non-compliance). Even sickle cell trait only provides around 50% protection against uncomplicated malaria and as such is less protective than an effective vaccine. These points are already discussed in the 5th paragraph of the discussion. While we appreciate the

reviewer's thoughts, therefore, we do not see how they apply to the case in hand and have not amended our manuscript in this regard. We would be happy to be persuaded to incorporate discussion of this point if considered necessary in the opinion of the editors.

Reviewer #2

1) This remains a potentially important paper. And several concerns have been partially addressed. I personally found the Mendelian randomization framework useful, but the other Reviewer appears to have objected.

No comment.

2) Re: selection of appropriate controls, the authors appear to have mitigated this major risk with the application of more germane controls. However, the description of them continues to confuse. Lines 304-307 indicate that "children aged 3-12 months of age who were born within the same study area as cases between August 2006 and September 2010... were used as controls," but then lines 311-315 state that "we constructed a control panel consisting of unselected children... between September 1998 and November 2005..." Which is it? Were there two control panels?

We apologise for this typographical error and thank the reviewer for pointing this out. The original controls were replaced by the alternative controls, but we omitted to delete the sentence on lines 304-307 from the amended version. We have now done so.

3) Table 3 can still be clarified by n/N that were used to compute at least the unadjusted ORs. I note that reviewer 1 also commented on the lack of clarity in Table 3, which I don't think has been improved substantially by the footnote. This transparency will help readers understand what is really the meat of the analysis, and enable meta-analyses by future groups, if people want to do that sort of thing. Even if it is derivable from elsewhere in the manuscript, I don't see a barrier to including them here.

We have amended Table 3 to include a column that shows the proportion of HbAS among case groups and have amended the footnote to state the proportion in controls and to drop some of the previous additions.

4) Lines 122-124. In this paragraph you have moved on to describing Table 3 results, but then in these lines you revert back to the prior analysis of case children only, which are results that you presented in a prior paragraph. Why go back?

While we acknowledge this point, the statements made are part of the logical flow of the story. We have considered re-sequencing this passage but have failed to find a better way to incorporate the results highlighted on lines 122-124. We would be happy to consider further if considered editorially essential.

5) Lines 131-134. This may be a question of journal policy, but I don't think it is good form to describe statistically insignificant findings as "marginal." It would be more appropriate to describe them as "decreases in risk that were not statistically significant" (irrespective of clinical significance).

We appreciate this point and have consequently deleted the sentence on lines 131-134 in the light of the reviewer's comment. The data are visible in the supplementary table and the deletion of this sentence makes no material difference to the manuscript.